

# A simple diagnostic based on sea surface height with application to Central Pacific ENSO

Jufen Lai[1,2,4], Richard J. Greatbatch[2,3], and Martin Claus[2,3]

[1]State Key Laboratory of Numerical Modeling for Atmospheric Sciences and Geophysical Fluid Dynamics, Institute of Atmospheric Physics, Chinese Academy of Sciences, Beijing, China
[2]GEOMAR Helmholtz Centre for Ocean Research Kiel, Germany
[3]Faculty of Mathematics and Natural Sciences, Christian Albrechts Universität, Kiel, Germany
[4]College of Earth and Planetary Sciences, University of Chinese Academy of Sciences, Beijing, China

**Correspondence:** Richard Greatbatch (rgreatbatch@geomar.de)

**Abstract.** We use output from a freely-running NEMO model simulation for the equatorial Pacific to investigate the utility of linearly removing the local influence of vertical displacements of the thermocline from variations in sea surface height. We show that the resulting time series of residual sea surface height, denoted $\eta_{nlti}$, measures variations in near-surface heat content that are independent of the local vertical displacement of the thermocline and can arise from horizontal advection, surface heat flux and diapycnal mixing processes. We find that the variance of $\eta_{nlti}$ and its correlation with sea surface temperature, are focused on the Niño4 region. Furthermore, $\eta_{nlti}$ averaged over the Niño4 region is highly correlated with indices of Central Pacific El Niño Southern Oscillation (CP ENSO), and its variance in 21 year running windows shows a strong upward trend over the past 50 years, corresponding to the emergence of CP ENSO following the 1976/77 climate shift. We show that $\eta_{nlti}$ can be estimated from observations, using satellite altimeter data and a linear multi-mode model. The time series of $\eta_{nlti}$, especially when estimated using the linear model, show pronounced westward propagation in the western equatorial Pacific, arguing an important role for zonal advective feedback in the dynamics of CP ENSO, in particular for cold events. We also present evidence that the role of the thermocline displacement in influencing sea surface height increased strongly after 2000 in the eastern part of the Niño4 region, at a time when CP ENSO was particularly active. Finally, the diagnostic is easy to compute and can be easily applied to mooring data or couple climate models.

## 1 Introduction

Coupling between the ocean and the atmosphere is the essential ingredient in the dynamics of El Niño Southern Oscillation (ENSO), as first pointed out by Bjerknes (1966) and as demonstrated by Philander et al. (1984) by coupling two shallow water models, one for the atmosphere, following Gill (1980), based on dynamics developed by Matsuno (1966), and one for the ocean, following Busalacchi and O'Brien (1981). The simple coupled system considered by Philander et al. (1984) exhibits disturbances that amplify and propagate eastward along the equator. A key feature of the model of Philander et al. (1984) is that the authors parameterised sea surface temperature (SST) anomalies as being proportional to the displacement of the thermocline, upward/downward displacement being associated with cold/warm SST anomalies. This simple idea is the





basis of the thermocline feedback (Jin et al., 2006) on SST that forms part of the Bjerknes feedback (Bjerknes, 1966, 1969) that has become central to our understanding of ENSO. Another important aspect of the Bjerknes feedback is the so-called
zonal advection feedback (Jin et al., 2006) in which SST anomalies are driven by anomalous zonal advection acting on an existing SST gradient. This mechanism was first investigated by Rennick (1983) and Rennick and Haney (1986) and shown by Hirst (1986) to lead to amplifying disturbances that propagate westward. It is generally understood that the thermocline feedback is the key mechanism for driving SST variability in the eastern equatorial Pacific, e.g. Zhang and McPhaden (2010), whereas zonal advection feedback is more important in the central, equatorial Pacific, e.g. Dewitte et al. (2013), and that both
thermocline and zonal advection feedback play a role in the evolution of ENSO events (Jin and An, 1999; An et al., 1999; An and Jin, 2001).

Rather than focus on SST, here we focus on anomalies in sea surface height (SSH) and note that because of the hydrostatic balance, variations in SSH are strongly influenced by local, vertical displacements of the thermocline. Linearly removing this influence leads to a time series for residual sea surface height that is dominated by changes in near-surface heat content arising
from variations in horizontal advection, surface heat flux and diapycnal mixing (see, for example, Hummels et al. (2014)). We illustrate the usefulness of this approach using output from a freely running ocean model simulation that includes the equatorial Pacific. The background is the emergence of Central Pacific (CP) ENSO since the 1976/77 climate shift (Larkin and Harrison, 2005; Ashok et al., 2007; Ren and Jin, 2011; Trenberth et al., 2002), a comprehensive review of which has been given by Capotondi et al. (2015). It turns out that the new approach is especially relevant for understanding the dynamics of CP ENSO,
as shown in Sect. 3. The diagnostic we propose is easy to compute and we suggest that it could be applied directly to mooring data or to coupled climate models. Such models are known to have problems simulating CP ENSO (Capotondi et al., 2015) and the diagnostic introduced here could provide insight into why this is the case.

The structure of the paper is as follows. Section 2 describes the methodology, beginning with the models and datasets used for analysis and then giving the theoretical background underpinning our approach. Section 3 presents the results and Sect. 4
provides a Summary and Discussion.

## 2   Methodology

### 2.1   Models and datasets used for analysis

As our basic dataset, we take output from a model for the global ocean driven by a time series of forcing based on observations from 1958 - 2019. The ocean model is the Nucleus for European Modelling of the Ocean (NEMO) (Madec et al., 2017), run at
a nominal horizontal resolution of $1/4° \times 1/4°$ with 46 vertical levels, and is the model configuration ORCA025.L46-KFS006 described in Biastoch et al. (2021) where more details can be found. The atmospheric forcing is provided by the JRA55-do dataset described by Tsujino et al. (2020).

In Sect. 3.2, we show how, using only knowledge of the surface wind stress, variations in the depth of the thermocline can be estimated and removed from variations in sea surface height, as measured, for example, by the satellite altimeter. To do this
we make use of the linear multi-mode model described by Zhu et al. (2017). In comparison with the NEMO model, the linear





model is easy to set-up and run, and is based on the early work of Busalacchi and O'Brien (1981). The model solves the linear shallow water equations forced by surface wind stress anomalies for the first several baroclinic vertical normal modes (Gill and Clarke, 1974; Gill, 1982), with the weighting given to each mode being determined by fitting, along the equator, model-computed SSH to SSH anomalies from an external dataset. We utilise two versions of the linear, multi-mode model (see Table 1), the first version being used to verify the method and the second version using only observational data sets. As such, the first version uses vertical modes based on the density stratification of the NEMO model simulation in the equatorial Pacific and model-computed SSH is fitted to SSH from the NEMO model simulation. The second version uses the same vertical modes as Zhu et al. (2017), computed using hydrographic data from the World Ocean Atlas (Locarnini et al., 2013), and model-computed SSH is fitted to AVISO satellite altimeter data (available from http://www.aviso.altimetry.fr/duacs/). Both versions are driven by the time series of monthly mean wind stress anomalies output from the NEMO model and derived from the JRA55-do dataset. When using SSH from the NEMO model as the external dataset (version 1), only the first three baroclinic modes can be separated and only these three modes are used. We also use only the first three baroclinic modes when fitting SSH from AVISO (version 2). As we show in Sect. 3, SSH from the linear model turns out to be a good surrogate for variations in the depth of the thermocline and, in fact, contrary to what one might think, the linear model is better at reproducing variations in the depth of the thermocline than it is at reproducing variations in SSH, an issue discussed further in Sect. 3.2.

**Table 1.** The two versions of the linear multi-mode model. NEMO refers to the NEMO model simulation, JRA55-do to the forcing product used to drive the NEMO model (see text for details), WOA to the World Ocean Atlas and SSH to the sea surface height product used for fitting in order to determine the weight given to each vertical mode.

| model | wind stress | stratification | SSH |
|---|---|---|---|
| Version 1 | JRA55-do | NEMO | NEMO |
| Version 2 | JRA55-do | WOA | AVISO |

It should be noted that the results shown in Sect. 3 are computed using monthly mean anomalies from a monthly mean climatology for the analysis period 1958-2019, apart from when using version 2 of the linear multi-mode model for which the analysis period is 1993-2019 when AVISO satellite data are available. All time series have been linearly detrended prior to the analysis and the significance levels given for the correlations shown in Sect. 3 are computed taking full account of the autocorrelation of the time series. We also use the Hadley Centre Sea Ice and Sea Surface Temperature data set (HadISST) (Rayner et al., 2003) as an independent reference with which to compute the indices based on SST shown in Fig. 4 and in Table 2.

## 2.2 Theoretical background

We take as our point of reference the two layer ocean shown in Fig. 1. The lower layer has density $\rho_2$ and the upper active layer has time mean depth $H_1$ and density $\rho_1 + \rho'$ where $\rho_1$ is the time mean density and $\rho'$ is the perturbation density associated with fluctuations about that state which, since we work in the near-surface equatorial ocean, are dominated by changes in potential





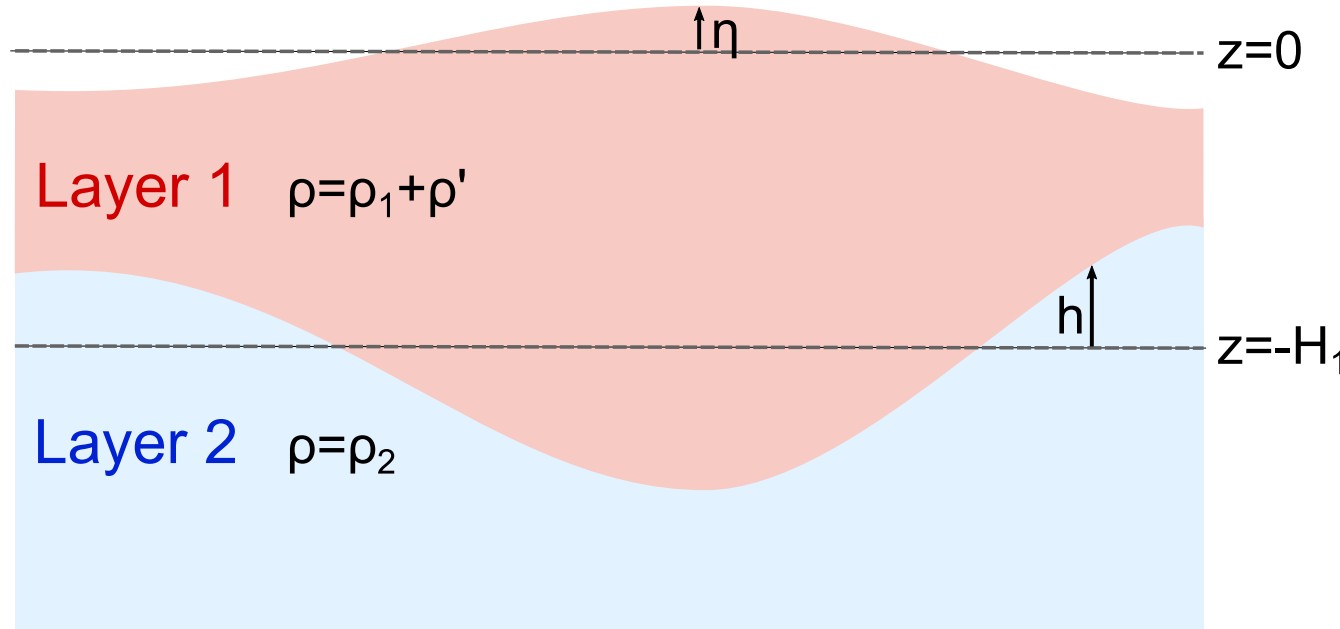

**Figure 1.** A schematic of the 2 layer model used in the theoretical background.

temperature. $\eta$ is the upward displacement of the sea surface and $h$ is the upward displacement of the interface between the two layers, both also associated with the fluctuations about the mean state. Integrating the hydrostatic balance upwards from a depth $-H_2$, where the pressure is time-independent and equals $p_2$, gives

$$85 \quad p_a - p_2 = -g \int_{-(H_1-h)}^{\eta} (\rho_1 + \rho')dz - g \int_{-H_2}^{-(H_1-h)} \rho_2 dz \tag{1}$$

where $p_a$ is atmospheric pressure (assumed to be time independent), $g$ is the acceleration due to gravity, and $z$ is distance measured positive upwards. Assuming for simplicity that $\rho_1$ and $\rho_2$ are independent of depth, linearising about the time mean state, removing the time mean, and rearranging gives

$$\eta = -h\frac{g'}{g} - \frac{1}{\rho_1} \int_{-H_1}^{0} \rho' dz. \tag{2}$$

90   $g' = g\left[\frac{\rho_2 - \rho_1}{\rho_1}\right]$ is the reduced gravity and depends on the density difference between the two layers and so is a measure of the strength of the stratification, an issue we return to later. Putting $\rho' = 0$, we recover the so-called 1 1/2 layer model, corresponding to an active upper layer and a deep, quiescent lower layer. In the 1 1/2 layer model, a downward/upward displacement of the thermocline is associated with an upward/downward displacement of the free surface at the same location, but of much smaller magnitude given by the ratio $g'/g$ - see Gill (1982). The difference here is the additional term $\frac{1}{\rho_1} \int_{-H_1}^{0} \rho' dz$ which,

95   since density is dominated by temperature, effectively measures perturbations to the heat content in the upper layer arising



from changes to the potential temperature of that layer. While this may also depend on $h$, it can also be influenced by other processes such as horizontal advection, the surface buoyancy flux and diapycnal mixing processes that are independent of $h$.

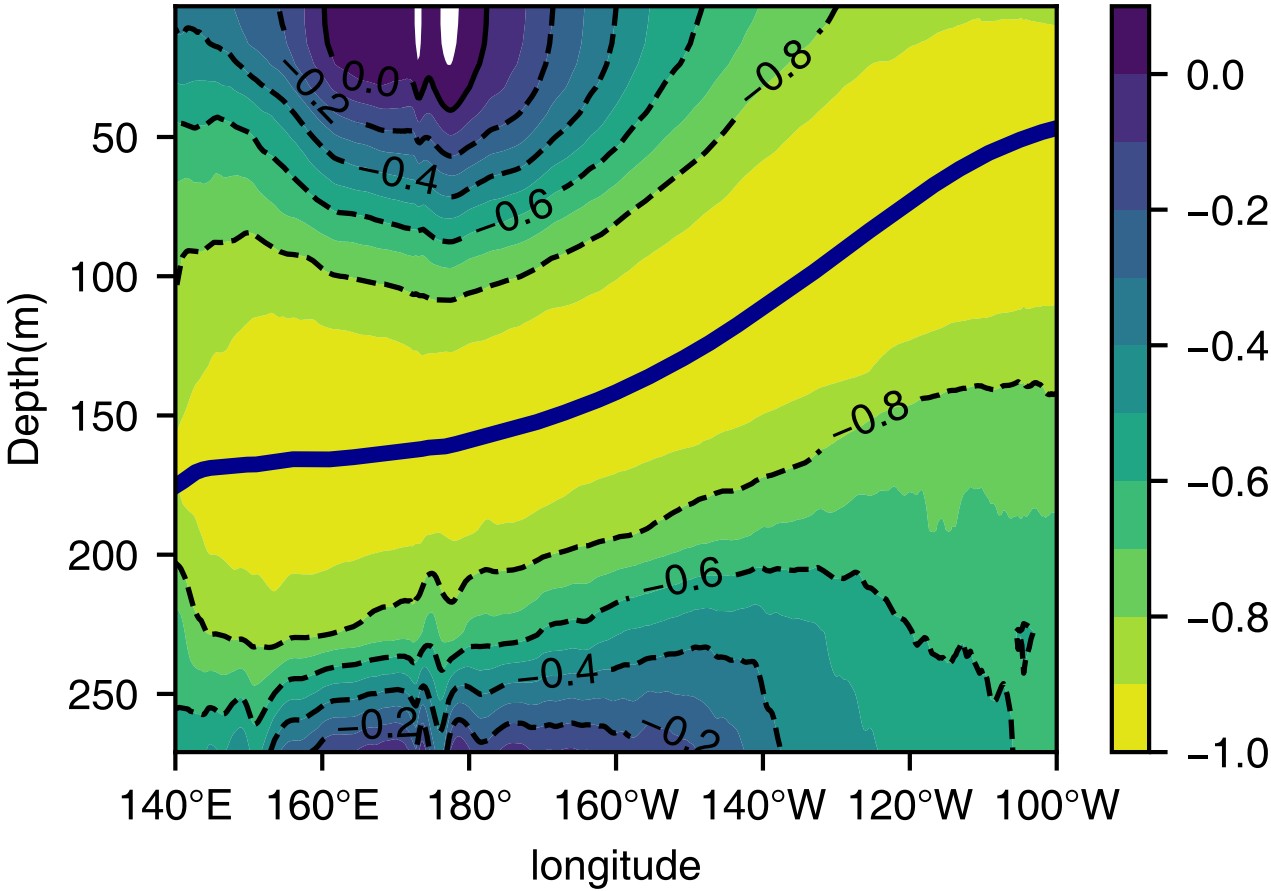

**Figure 2.** The correlation along the equator between anomalies in D20 and potential temperature in the NEMO model simulation. The mean depth of D20 is shown by the thick, dark blue line. Note that since we are at the equator, variations in density are determined mostly by variations in potential temperature.

We follow Dewitte et al. (2013), and use anomalies in the depth of the $20°C$ isentrope (hereafter D20) as a measure for variations in the depth of the thermocline, corresponding to $h$ in Eq. (2). In doing so, we are assuming that the dominant variability takes place in association with vertical displacements of D20, and between D20 and the sea surface. We use linear regression to remove the influence, at each location separately, of variations in D20 from variations in sea surface height (hereafter SSH), corresponding to $\eta$ in Eq. (2). The resulting time series of residual sea surface height is referred as $\eta_{nlti}$,



where $nlti$ refers to "No Linear Thermocline Influence", is then given by

$$\eta_{nlti} = \eta - \alpha h. \tag{3}$$

Here $h$ refers to departures in D20 from the mean, and $\alpha$ is the coefficient obtained by linear regression of $h$ on $\eta$. Based on the simple model described by Eq. (2), the regression coefficient, $\alpha$, contains information on the strength of the stratification through the dependence on $g'/g$ and it is, therefore, important to carry out the regression at each location separately, in order to take account of horizontal variations in the stratification. The regression analysis removes the (major) effect on SSH from variations in the depth of the thermocline, in particular coming from the first term on the right hand side of Eq. (2), but leaves the impact on SSH from other, independent processes. Looking at Fig. 2, it can be seen that between $150°$E and $160°$W, the variations in density above the $20°C$ isentrope are much more weakly correlated with variations in D20 than they are to either the east or the west along the equator. It is, therefore, in this region that, in the context of Eq. (2), we would expect the term $\frac{1}{\rho_1} \int_{-H_1}^{0} \rho' dz$ to play a role, and hence $\eta_{nlti}$ to exhibit significant variance. It should be noted that the region between $150°$E and $160°$W along the equator corresponds closely to the Niño4 region (see https://climatedataguide.ucar.edu/climate-data/nino-sst-indices-nino-12-3-34-4-oni-and-tni) and that this is also the region where SST anomalies associated with CP ENSO reach their peak. In Sect. 3 we examine both the spatial distribution of the variance of $\eta_{nlti}$ and the relationship between $\eta_{nlti}$ and SST in the equatorial region.

Part of the original motivation for working with $\eta_{nlti}$ came from Fig. 2 in Dewitte et al. (2013). This figure is computed using the SODA reanalysis dataset described by Giese and Ray (2011). The close similarity between the lower panel in that figure (which, in fact, shows correlation and not the regression coefficient as stated in the caption) and our Fig. 2 provides an independent verification of the freely running NEMO model simulation we use for analysis. It should be noted that because Dewitte et al. (2013) measure D20 positive downwards and we measure D20 positive upwards, there is a change of sign between the two figures.

## 3 Results

### 3.1 The relationship to SST and CP ENSO

Figure 3a shows the variance of monthly mean anomalies of SST, D20 and $\eta_{nlti}$ along the equator in the Pacific sector of the NEMO model simulation. It is striking how the variance of both SST and D20 increase eastwards along the equator. Also notable is the low level of variance of $\eta_{nlti}$ in the eastern equatorial Pacific, and the peak in the variance of $\eta_{nlti}$ in the Niño4 region around the date line ($180°E$) that was anticipated when we discussed $\eta_{nlti}$ in the context of Fig. 2. Figure 3b shows the correlation between monthly mean anomalies of SST and D20, and SST and $\eta_{nlti}$. The strong influence of D20 on SST in the eastern equatorial Pacific is evident, as is the weak positive correlation between SST and D20 between $160°E$ and the date line ($180°E$). The latter corresponds to the weak positive correlation in this region in Fig. 2 and the weak negative correlation in Fig. 2 of Dewitte et al. (2013), and also at zero lag in Fig. 2 of Zelle et al. (2004), noting that we are measuring D20 positive upwards whereas these authors measure D20 positive downwards. On the other hand, the correlation between SST and $\eta_{nlti}$ is





**Figure 3.** (a) The variance of detrended time series of monthly mean anomalies of SST (blue), $\eta_{nlti}$ (orange), and D20 (dashed) as a function of longitude along the equator in the NEMO model simulation. (b) Same as (a) but for the correlation of the different indices with SST. The units in (a) are $(^\circ C)^2$ (left hand scale) for SST; $m^2$ (right hand scale) for D20 and $\eta_{nlti}$. Correlations in (b) that are outside the grey, shaded region are significantly different from zero at the 95% level.





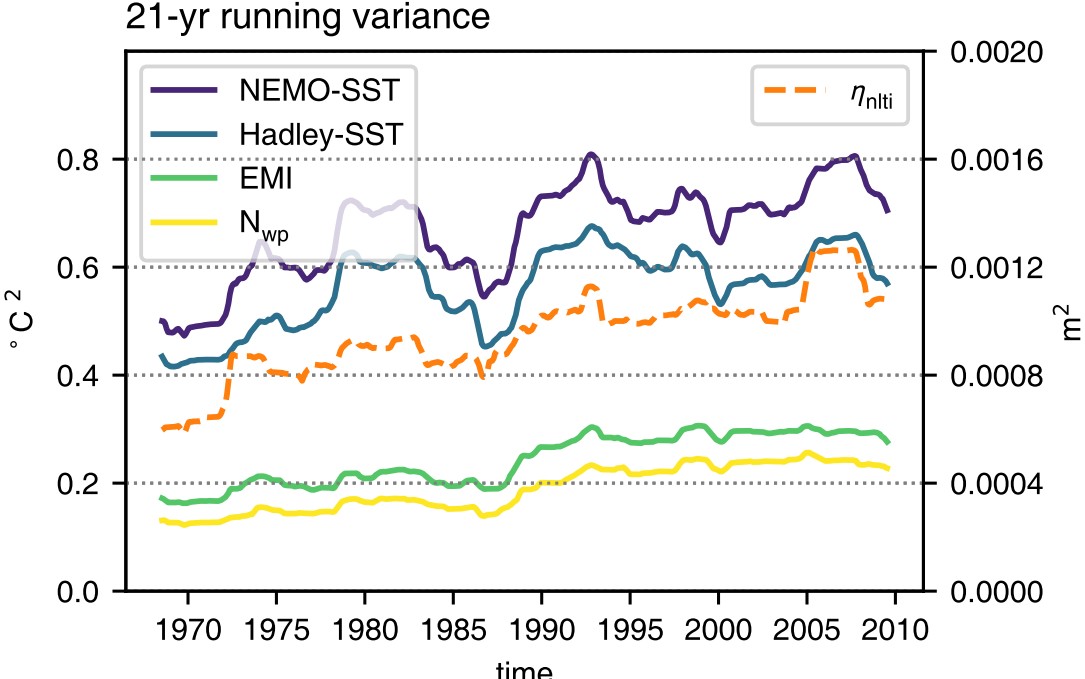

**Figure 4.** Variance in running 21 year windows of time series of different indices averaged along the equator between $160°E$ and $150°W$, corresponding to the Niño4 region and computed using monthly mean anomalies. NEMO-SST refers to SST taken from the NEMO model simulation and Hadley-SST to SST taken from the HadISST data set. The units are $(°C)^2$ (left axis) except for the index based on $\eta_{nlti}$ which has units $m^2$ (right axis).

135 close to zero in the eastern equatorial Pacific but reaches above $0.8$ in the Niño4 region between $160°E$ and $150°W$. In fact, the correlation between SST and $\eta_{nlti}$, after both variables are detrended and averaged over the Niño4 region, including off the equator, is $0.92$ (see Table 2).

 We noted when discussing Fig. 3 that most of the variance in $\eta_{nlti}$, and also the high correlation with SST, is confined to the western, central equatorial Pacific where CP ENSO is active. Table 2 shows the correlation between $\eta_{nlti}$ and SST, both

140 averaged over the Niño4 region, and two indices that have been used for CP ENSO, the El Niño Modoki Index (EMI) from Ashok et al. (2007) and the Warm Pool ($N_{WP}$) index of Ren and Jin (2011). From the table, one can see that $\eta_{nlti}$ is highly correlated (up to $0.9$) with all of Niño4 SST, EMI and $N_{WP}$. The high correlation with $\eta_{nlti}$ indicates the importance of processes, independent of the local thermocline displacement, in the dynamics of CP ENSO. The strong connection between CP ENSO and $\eta_{nlti}$ is also apparent from Fig. 4 which shows the running variance of the different indices in 21 year windows.

145 The upward trend in running variance in all the indices is symptomatic of the increasing importance of CP ENSO since the



1976/77 climate shift (Ashok et al., 2007; Ren and Jin, 2011; Lübbecke and McPhaden, 2014). It is remarkable that the variance of the $\eta_{nlti}$-based index effectively doubles over the analysis period.

**Table 2.** Correlation between detrended time series of different indices for CP ENSO. The SST indices, and $\eta_{nlti}$, are averaged over the Niño4 region. The upper table uses SST from the NEMO model simulation and the lower table uses SST from HadISST. All correlations are significantly different from zero at the 99% level.

| NEMO | $\eta_{nlti}$ | $N_{WP}$ | EMI |
|---|---|---|---|
| SST | 0.92 | 0.88 | 0.90 |
| $\eta_{nlti}$ | | 0.87 | 0.88 |
| $N_{WP}$ | | | 0.94 |

| HadISST | $\eta_{nlti}$ | $N_{WP}$ | EMI |
|---|---|---|---|
| SST | 0.88 | 0.85 | 0.77 |
| $\eta_{nlti}$ | | 0.81 | 0.75 |
| $N_{WP}$ | | | 0.93 |

### 3.2 Estimating $\eta_{nlti}$ from observations

We now show how to estimate $\eta_{nlti}$ from observations. To do this, we make use of the linear multi-mode model described by Zhu et al. (2017). We begin with Version 1 in Table 1 in order to verify the method, and then discuss Version 2 that uses only data based on observations. Both versions are driven by the time series of monthly mean wind stress anomalies that are output from the NEMO model simulation and are derived from the JRA55-do dataset (Tsujino et al., 2020). We begin with the first version. This uses vertical modes computed from the density stratification of the NEMO model with the weighting given to each mode set by fitting, along the equator, SSH output from the linear model to SSH anomalies from the NEMO model. Only the first three baroclinic modes are used (adding more modes does not improve the fit). Figure 5a shows the correlation between the monthly means of SSH from the linear model and monthly mean anomalies of both SSH and D20 (sign reversed) from the NEMO model as a function of longitude along the equator and for the whole time period (1958-2019) covered by the NEMO model simulation. The drop in correlation between the two SSH products in the Niño4 region is clearly visible and was also noted by Zhu et al. (2017) when comparing their model output with AVISO data. On the other hand, this drop in correlation is much less noticeable when SSH from the linear model is correlated with D20 from the NEMO model, arguing that the linear model of Zhu et al. (2017) is really a model for anomalies of D20 and not SSH. The lower panels in Fig. 5 show Hovmoeller diagrams along the equator of anomalies of SSH and D20 (sign reversed) from the NEMO model (Figs. 5b,c) and SSH from the linear model (Fig. 5d). From this figure it is clear that the features in SSH from NEMO (Fig. 5b) are shifted further west compared to those in either D20 from NEMO (Fig. 5c) or SSH from the linear model (Fig. 5d). This westward displacement was noted by Zhu et al. (2017) and is clearly the reason for the drop in correlation in the Niño4 region between SSH from the linear model and SSH from NEMO (or indeed from AVISO, as discussed by Zhu et al. (2017)). Based on the above analysis,



**Figure 5.** (a) The correlation between SSH from version 1 of the linear model and anomalies of SSH (blue) and D20 (sign reversed) from the NEMO model simulation (black dashed). All time series have been detrended and all the correlations shown are significantly different from zero at the 99% level. Also shown are Hovmoeller diagrams (units of metres) along the equator of anomalies of (b) SSH and (c) D20 from the NEMO model and (d) SSH from version 1 of the linear model. Note that the sign of the D20 anomalies have been reversed for ease of comparison in panel (c).







**Figure 6.** Hovmoeller diagrams as a function of longitude along the equator for (a) $\eta_{nlti}$ computed from the NEMO model simulation and (b) $\eta_{nlti}$ computed using version 1 of the linear multi-mode model. The units are metres.



**Figure 7.** Hovmoeller diagrams (units of metres) along the equator of (a) $\eta_{nlti}$ computed from the NEMO model simulation, (b) $\eta_{nlti}$ computed using version 1 of the linear model using SSH from the NEMO model, and (c) $\eta_{nlti}$ computed using version 2 of the linear model using AVISO data.

we now use SSH from the linear model as a surrogate for D20. $\eta_{nlti}$ is then computed using (i) SSH from the linear model in place of D20 and (ii) SSH anomalies taken from the NEMO model simulation for SSH. The result is shown in Fig. 6b for comparison with Fig. 6a where $\eta_{nlti}$ is computed using both D20 and SSH taken from the NEMO model simulation. The high

level of agreement between these two figures indicates that using SSH from the linear multi-mode model, as a surrogate for variations in D20, is a viable option that can, in turn, be used to estimate $\eta_{nlti}$ based on observed estimates of surface wind stress as forcing for the linear model, and AVISO satellite data for SSH.

To illustrate this, we now use Version 2 of the linear multi-mode model (see Table 1), this time based on vertical normal modes computed from the World Ocean Atlas and with the weighting given to each mode set by fitting, along the equator, SSH





output from the linear model to AVISO data along the equator. $\eta_{nlti}$ is then computed using SSH from the linear model as a surrogate for anomalies in D20, and AVISO satellite data for anomalies in SSH. The result is shown in Fig. 7c and compared with $\eta_{nlti}$ from the NEMO model simulation (Fig. 7a) and version 1 of the linear model (Fig. 7b) that is based on output from the NEMO model simulation. The results shown in Fig. 7 cover the period 1993-2019 for which AVISO data are available; Fig. 6a,b are the same as Fig. 7a,b, respectively, but for the extended period 1958-2019 covered by the NEMO simulation. It is

important to note that what is shown in Fig. 7c is based solely on observations; it is not a diagnostic from a model simulation, as is the case for the time series shown in Fig. 7a. Nevertheless, the different time series shown in Fig. 7 are clearly very similar to each other, be there some minor differences; in particular, the major El Niño event of 1997/98 appears stronger in Fig. 7a than in Figs. 7b,c that use the linear model, and the variance is clearly higher in the east when AVISO data is used for SSH (Fig. 7c). What is interesting, however, is that there is often westward propagation in the western basin, especially for features that appear

in blue, i.e. cold events. For example, in 2009, there is an event for which the propagation speed is close to $0.25 \text{ m s}^{-1}$ in Figs. 7b,c, be it a little slower in the time series of $\eta_{nlti}$ from the NEMO model simulation (Fig. 7a). The less prominent westward propagation in Fig. 7a may possibly arise from the impact of advection on D20 by the eastward Equatorial Undercurrent that is missing from the linear model. Since using the linear model to estimate D20 excludes this advective effect, the resulting diagnosis of $\eta_{nlti}$ reflects processes taking place in the water column above the $20°C$ isentrope and, in particular, interaction

with the atmosphere. As such, the indication of westward propagation suggests that coupled air-sea interaction associated with the zonal advection feedback, and similar to that noted by Hirst (1986), may be active in the western basin in the observations.

## 4 Summary and discussion

We have shown the utility of a simple diagnostic for the interpretation and analysis of data from the equatorial ocean. The new diagnostic, denoted here as $\eta_{nlti}$ ($nlti$: No Linear Thermocline Influence), is defined at each location as the time series

of anomalies in sea surface height (SSH) with the effect of anomalies in the depth of the $20°C$ isentrope (D20; anomalies measured positive upward) removed by linear regression, the latter being performed separately at each location to account for spatial inhomogeneity in the stratification. The time series of $\eta_{nlti}$ measures changes in near-surface heat content that are independent of local, vertical displacements of the thermocline and can arise from horizontal advection, surface buoyancy flux and diapycnal mixing processes. We have shown that the variance of $\eta_{nlti}$ is concentrated in the Niño4 region and that, after

averaging over the Niño4 region, $\eta_{nlti}$ is highly correlated with SST and indices used to monitor CP ENSO. Furthermore, the running variance of $\eta_{nlti}$ (averaged along the equator in the Niño4 region) in 21 year running windows (Fig. 4) shows a remarkable upward trend that is consistent with the emergence of CP ENSO after the 1976/77 climate shift (Ashok et al., 2007; Ren and Jin, 2011) and the relatively high frequency of CP ENSO events after 2000 (Lübbecke and McPhaden, 2014). Interestingly, the time series of running variance (Fig. 4) shows no influence of the Pacific Decadal Oscillation (PDO) (Mantua

et al., 1997), consistent with Lübbecke and McPhaden (2014) who have previously noted that the occurrence of CP ENSO appears to be independent of the phase of the PDO. We have suggested that the dynamics of CP ENSO, especially cold events, are related to the coupled atmosphere-ocean feedback identified by Hirst (1986), in which SST anomalies are generated by




anomalies in zonal advection, i.e the so-called zonal advection feedback. Of particular interest is the westward propagation in the western basin evident from Figs. 6 and 7 supporting this view.

We have also shown that the linear, multi-mode model of Zhu et al. (2017) should be interpreted as a model for variations in D20 and not SSH. Indeed, such a model, driven by a time series of observed wind stress anomalies, generates a time series of SSH that can then be used as a surrogate for anomalies of D20. Combining this time series with satellite altimeter data then gives a time series of $\eta_{nlti}$ based on observations.

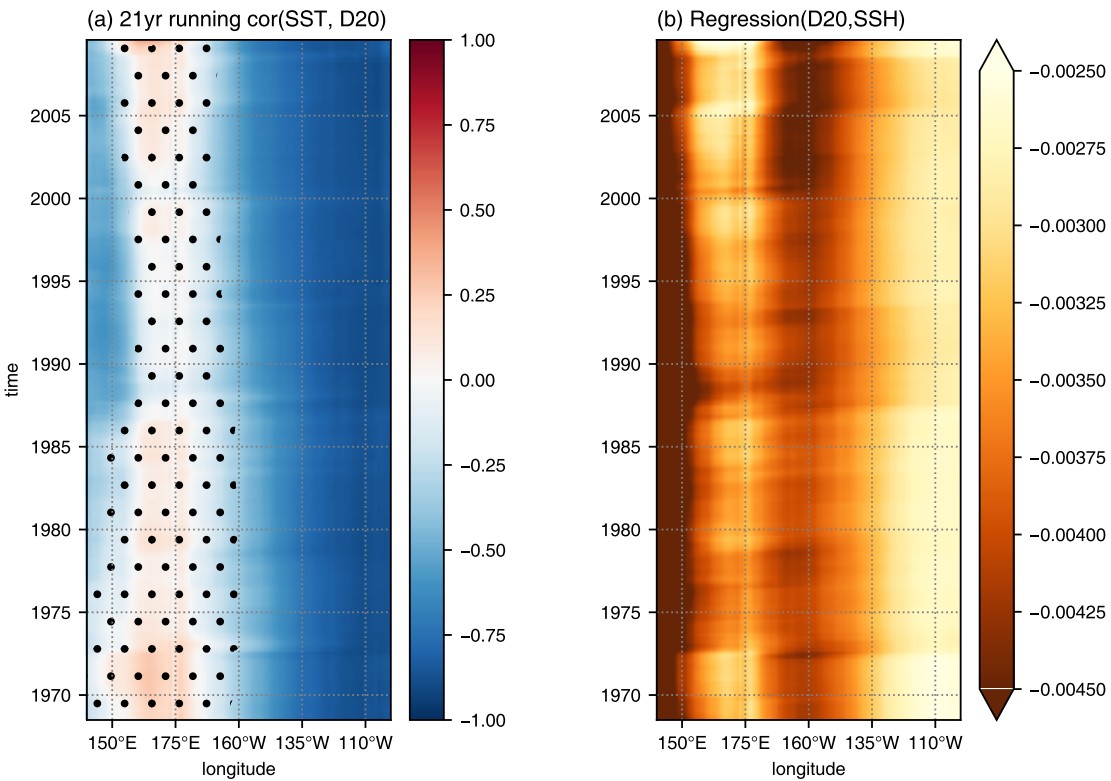

**Figure 8.** (a)The correlation in running 21 year windows between anomalies of SST and D20 from the NEMO model simulation. The time series have been detrended in each window separately. Dots indicate regions where the correlation is not significantly different from zero at the 95% level. (b) The coefficient, in running 21 year windows, obtained by regressing D20 on sea surface height from the NEMO model simulation. The time series have been detrended in each window separately. The regression slopes are everywhere significantly different from zero at the 95% level.

There has been much discussion on the possible role of thermocline feedback in the emergence of CP ENSO (Yeh et al., 215    2009; Dewitte et al., 2013; Greatbatch et al., 2018; Zhu et al., 2021). The correlation in 21 year running windows between anomalies of SST and D20 (a simple measure of thermocline feedback) is shown in Fig. 8a from the NEMO model simulation. Over most of the Niño4 region, between 160°E and 150°W, the correlation is always close to zero, although there is tendency





for the significant negative correlations that dominate the eastern basin to encroach westwards into the Niño4 region over time. Also of interest is the emergence of a region of significant negative correlation to the west of the Niño4 region in the windows

centred on the mid-1980's. For comparison, Fig. 8b shows a Hovmoeller diagram along the equator of the regression coefficient (that is $\alpha$ in Eq. (3)) obtained by regressing anomalies of D20 against anomalies of sea surface height from the model in 21 year running windows (the time series are detrended separately in each window). As noted following Eq. (2), this regression coefficient (which is negative) measures the importance of processes related to the local thermocline displacement and can be interpreted as a measure of the strength of the stratification when the first term on the right hand side of Eq. (2) is dominant.

Figure 8b clearly shows a trend in the model simulation, especially in the eastern part of the Niño4 region west of the date line, with a sharp increase in the importance of thermocline displacement after 2000 coinciding with the time when CP ENSO became most active (Lübbecke and McPhaden, 2014). What role this trend plays in the dynamics of CP ENSO remains a topic for future research, but it seems clear that processes related to the local thermocline displacement became more important after 2000.

When discussing Fig. 3, we ignored the seasonal cycle in the correlation between anomalies in thermocline depth and SST (Zhu et al., 2015). However, a preliminary analysis shows that treating each month separately and repeating the analysis presented here has no significant effect on the results (not shown).

    Finally, the new diagnostic can be easily calculated from either model output or observations, including directly from mooring data, and can be easily applied to other equatorial ocean basins, as will be shown elsewhere. As noted in the introduction,

applying the diagnostic to coupled climate models could provide insight into why these models have problems simulating CP ENSO (Capotondi et al., 2015).

*Code and data availability.* Datasets for this research are available from Claus et al. (2022)[CC-by 4.0]. Software for this research is available from Claus and Lai (2022)[BSD License].

*Author contributions.* All authors contributed to the conceptualisation and methodology and JL carried out the formal analysis with help
from MC. RJG prepared the manuscript with contributions from all co-authors.

*Competing interests.* The authors declare that they have no conflict of interest.

*Acknowledgements.* Jufen Lai acknowledges funding from the CAS-DAAD Doctoral Joint Scholarship Program, in particular the CAS-DAAD Regierungsstipendien - Programme for the Promotion of Outstanding Young Scholars, 2020 (57535487). We thank Franziska Schwarzkopf for providing the output from the NEMO configuration used in this study and Peter Brandt for his steadfast encouragement.



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
