# Peer review of "A simple diagnostic based on sea surface height with application to Central Pacific ENSO"

_EGUsphere, 2022_

## Author Comment (AC1)

We thank the reviewer for their constructive and helpful comments. In the following, the comments from the reviewer are shown using a blue, italic font and our response in black, regular font.

*Lai et al. (2023) developed a new index using sea surface height by removing the local thermocline feedback to revisit its relationship with SST variability over the Central Pacific region. The new index corresponds well to the CP ENSO SST index, confirming the importance of thermocline feedback to EP ENSO and zonal advection feedback to CP ENSO. While I found this is a good approach to test whether model simulations could simulate relevant dynamics associated with different ENSO regimes, some points could be made clearer*

*Major points:*

1. *3b shows the correlation of the new index with SST. I would suggest adding the correlation of the original SSH with SST. This will highlight the necessity of removing the local thermocline feedback.*

We now show the correlation of SSH with SST in Figure 3b and added a corresponding sentence to the text. The correlation of SST with SSH is close to one in the east and drops off to close to 0.5 in the west.

2. *The point made by this study that the zonal advection feedback is important for CP ENSO events especially for La Nina (Lines 184-185) is not clear from the current figures. Authors are encouraged to modify the Hovmoeller diagrams in Figs. 5-7 with arrows to indicate the propagations in CP ENSO years.*

We use arrows to highlight some of the events that exhibit westward propagation in what is now Figure 8c. We have also noted in the text that what are now Figures 7 and 8 capture documented CP ENSO events with reference to the appropriate papers.

3. *It is true that the (nonlinear) zonal advection term is particularly important to CP La Nina, leading to the negative SST skewness in CP region. Could authors plot the new index and generate its skewness?*

eta_nlti and SST averaged over the Nino4 region have negative skewness, as do the indices EMI and N_WP that have been used to measure CP ENSO. We now show histograms of the different indices in the new Figure 4 and also added a sentence to the main text on this topic.

4. *The method of removing the local thermocline feedback only considers the concurrent response. The reason for the negative relationship (D20 positive downward) over the CP region (Fig. 2) is that D20 leads the response of CP SST (Zelle et al. 2004; https://doi.org/10.1175/2523.1). Whether the current method can make it clean from the D20 influence is unknown. It should be discussed.*

An advantage of our approach is that it is based on the hydrostatic balance as described by our equations (1) and (2). This is an instantaneous balance and hence is concerned only with the concurrent response and not with lagged relationships. We now comment on this in the text.

*5. It would be better to test this method in one of the models which can generally simulate both CP and EP ENSO events (Fig. 4 in Cai et al. 2021 https://doi.org/10.1038/s43017-021-00199-z).*

We have decided not to do this. Rather we want to defer application to coupled models to a future study. The present manuscript is about verifying the utility of eta_nlti as a diagnostic using a freely running ocean model and against observations.

*Minors:*

*I found Fig.2 is confusing using D20 positive upward. I would suggest revising it following what most other studies would do.*

We now measure D20 positive downward throughout.

---

## Author Comment (AC2)

We are very grateful to the reviewer for their very positive assessment of our manuscript. Regarding the last point raised by the reviewer, we have done some preliminary analysis of the heat budget of the ocean-only GCM we use. However, the analysis is not easy to carry out and the results are not as insightful as obtained using the simple approach we adopt in the manuscript, the essence of which is, in any case, already contained in what is shown in Figure 2. Indeed, we think it best not to complicate matters by including such an analysis. The simplicity of our approach also makes it ideal for application to coupled models which we know have problems simulating CP ENSO. The diagnostic can also be applied to mooring data, an approach we have been exploring with our colleague Prof. Peter Brandt, a topic of ongoing research.

*Review of a manuscript entitled "A simple diagnostic based on sea surface height with application to Central Pacific ENSO" by Jufen Lai et al.*

*I have read through the manuscript with much interest. The authors have developed a simple method to derive near-surface heat content changes by taking advantages of sea-surface height anomalies and removing the contribution from the thermocline displacement. Then they have shown the importance of the thermal anomalies mostly due to horizontal advection, primarily in the evolution of the Central Pacific ENSO (or Modoki ENSO).*

*Thermodynamic (heat and moisture) and dynamical (momentum) fluxes at the sea surface are important for understanding the atmosphere-ocean interactions that produce climate variability modes such as ENSO. In particular, the thermodynamic effects that determine sea surface temperature are of great importance. Understanding the evolution of ENSO can be broadly divided into two approaches. One approach is that the zonal advection of the surface water temperature is important, and the other is that the influence of the thermocline on the surface temperature by oceanic mixed layer processes is important. The former is abbreviated as zonal advection feedback and the latter as thermocline feedback.*

*The thermocline variability in the tropical equatorial region is mainly due to dynamic forcing by momentum flux from the ocean surface, but thermocline feedback does not ignore thermodynamics. It implicitly assumes the oceanic mixed layer above the thermocline, just skipping the process for brevity. Naturally, there is a lag between thermocline variability and the sea surface temperature variability.*

*The authors have considered briefly the sea level variability in two parts, based on the two-layer model. One is caused by fluctuations in the thermocline and the other is caused by fluctuations in surface heat capacity. Such brevity is important for understanding physics. It may also open the door to exploiting data from altimeter satellites. I would like to commend the authors for taking on this challenge. The authors have attempted to interpret the ENSO diversity for concrete applications and have suggested that the Central Pacific ENSO (or Modoki ENSO) relies more on zonal advection feedback.*

*In this paper, by using seemingly dynamic fluctuations, the authors have derived differences in thermodynamic mechanisms of surface water temperature fluctuations in the present paper. Ideally, the authors could have used the results of GCM simulations to perform a thermodynamic analysis of fluctuations of sea surface temperature (or surface mixed layer temperature), and demonstrated the effectiveness and limitations of the simple approach introduced here. However, I think this work itself deserves publication.*